# Advances in Antigenic Peptide-Based Vaccine and Neutralizing Antibodies against Viruses Causing Hand, Foot, and Mouth Disease

**DOI:** 10.3390/ijms20061256

**Published:** 2019-03-13

**Authors:** Mohd Ishtiaq Anasir, Chit Laa Poh

**Affiliations:** Centre for Virus and Vaccine Research, Sunway University, Bandar Sunway, Subang Jaya, Selangor 47500, Malaysia; ishtiaqa@sunway.edu.my

**Keywords:** vaccine, antigenic peptide, neutralizing antibodies

## Abstract

Hand, foot, and mouth disease (HFMD) commonly produces herpangina, but fatal neurological complications have been observed in children. Enterovirus 71 (EV-A71) and Coxsackievirus 16 (CV-A16) are the predominant viruses causing HFMD worldwide. With rising concern about HFMD outbreaks, there is a need for an effective vaccine against EV-A71 and CV-A16. Although an inactivated vaccine has been developed against EV-A71 in China, the inability of the inactivated vaccine to confer protection against CV-A16 infection and other HFMD etiological agents, such as CV-A6 and CV-A10, necessitates the exploration of other vaccine platforms. Thus, the antigenic peptide-based vaccines are promising platforms to develop safe and efficacious multivalent vaccines, while the monoclonal antibodies are viable therapeutic and prophylactic agents against HFMD etiological agents. This article reviews the available information related to the antigenic peptides of the etiological agents of HFMD and their neutralizing antibodies that can provide a basis for the design of future therapies against HFMD etiological agents.

## 1. Introduction

Hand, foot, and mouth disease (HFMD) is a prevalent paediatric infectious disease that is caused by a variety of enteroviruses [1,2,3,4]. HFMD has become a major public health threat that leads to serious morbidity and mortality in Asia-Pacific. In 2017, a total of 1,952,435 HFMD cases were recorded in China [5]. Enterovirus 71 (EV-A71) and Coxsackievirus 16 (CV-A16) are the main HFMD causative agents of mild HFMD. However, some EV-A71 strains have been known to be associated with fatal neurological complications [1,2]. In recent years, many HFMD outbreaks are associated with emerging enteroviruses, including CV-A6, CV-A8, and CV-A10 [3,4,6]. Together with EV-A71 and CV-A16, these enteroviruses have been shown to co-circulate in many HFMD outbreaks, which may result in viral co-infections and genomic recombination [7,8].

With rising concern about HFMD outbreaks, there is an urgent need for a vaccine against EV-A71 to be produced that is approved by the United States Food and Drug Administration (FDA) or European Medicines Agency (EMEA). The inactivated EV-A71 vaccine has typically been used as the main strategy for vaccine design [9]. China’s FDA has issued drug certificates and production licenses for the inactivated vaccines from 3 companies, namely, Sinovac, Vigoo, and Chinese Academy of Medical Sciences (CAMS) [9]. Sinovac reported that the efficacy of their inactivated vaccine against EV-A71-associated HFMD cases was 94.8% during the 12-month surveillance period [10]. Besides that, CAMS and Vigoo reported 97.4% and 90.0% vaccine efficacy against EV-A71-associated HFMD, respectively [11,12]. Despite the promising results revealed by the safety and efficacy studies, the licensed vaccines are incapable of eliciting sufficient cross-neutralizing antibodies against the other HFMD causative agents, such as CV-A16, CV-A6 and CV-A10 [9]. Therefore, it is imperative to explore other vaccine strategies to produce a safe and effective vaccine against multiple HFMD etiological agents.

A comprehensive understanding of the antigenic peptides that could elicit neutralizing antibodies will greatly help in the design of an effective vaccine. We discuss the structural analysis of EV-A71 viral particles to identify possible foreign peptide insertion sites to develop a multivalent vaccine based on a virus-like particle (VLP) platform. Lastly, we suggest some strategies to develop multivalent vaccines against multiple viruses that can cause HFMD.

## 2. Etiological Agents of HFMD

HFMD etiological agents (EV-A71, CV-A16, CV-A6, and CV-A10) belong to the *Picornaviridae* family, within species A of the Enterovirus genus [13]. The Enterovirus A species possesses non-enveloped virus particles with symmetrical 20–30 nm icosahedral capsids. The viral particle contains a single-stranded positive sense 7.4 Kb ribonucleic acid (RNA) with a single open reading frame, which is flanked by a 5′-untranslated region (5′ UTR) and a 3′ UTR with a poly(A) tail. Enterovirus A encodes a polyprotein of approximately 2100 amino acids that will be further cleaved to form three precursor proteins, termed as P1, P2, and P3. Structural viral proteins, named as VP1, VP2, VP3, and VP4, are the products of P1 cleavage. Besides that, P2 and P3 cleavage form seven non-structural proteins, namely 2A, 2B, and 2C, which are derived from P2, and 3A, 3B, 3C, and 3D, which are derived from P3. Enterovirus A viral capsids are formed by 60 protomers of the structural viral proteins; each protomer consists of VP1, VP2, VP3, and VP4 [14]. The external capsid surface is formed by VP1, VP2, and VP3, while VP4 is located inside the capsid [14]. VP1, VP2, and VP3 are structurally similar to one another, each consisting of eight-stranded β-barrels.

Comprehensive structural and biochemical studies reported five recognizable viral particles throughout the Enterovirus A lifecycle. They are the putative pro-capsid, the pro-virion, the mature infectious virus, the “A-particle”, and the empty capsid. Particle assembly commences in the cytoplasm where the protomer is formed by viral structural proteins VP0, VP1, and VP3. Five protomers will then assemble to form the pentameric subunit, which ultimately self-assembles to form the 60 protomer pro-capsid. The pro-capsid will acquire the viral genome to form the pro-virion. After the formation of the pro-virion, VP0 will be autocatalytically cleaved into VP2 and VP4 to form the mature virions that are released from infected cells. The mature virions will bind to neighbouring host receptors on the cell surface and begin the two-step uncoating process, which are (1) the formation of an expanded and altered particle, termed as “A-particle”, which is primed for genome release, and (2) the release of RNA into the host cell generating an intact empty capsid. VP4 is essential during the uncoating process as it is inserted into the membrane to form a channel in which the RNA can be released into the host cell cytoplasm [15,16,17,18,19].

The mature virus particles and the empty capsid are predominantly produced during EV-A71 natural infection. These two particles can be separated using continuous sucrose gradient ultra-centrifugation. The empty capsids have ~5% larger diameter than the mature virus particles [19]. Crucially, both viral particles displayed distinct antigenic properties, whereby the empty capsids are hypothesised to act as baits for host immune evasion [20].

## 3. Neutralizing Epitope Mapping

### 3.1. EV-A71 Neutralizing Epitope Mapping

Over the years, a number of linear neutralizing epitopes in the enteroviruses causing HFMD have been reported by many research groups. These reported linear and conformational epitopes were mostly mapped to the enterovirus structural proteins VP1, VP2, VP3, and VP4. We have summarized the reported linear neutralizing epitopes from EV-A71 and CV-A16 in Table 1.

VP1 is a capsid protein comprising 297 amino acids and is located at the most external part of the viral capsid. VP1 was the first EV-A71 antigen found to contain important epitopes that contribute to the neutralization of the virus. Initially, Wu et al. (2001) found that immunization using a recombinant VP1 protein of EV-A71 expressed in *E. coli* conferred protection against lethal EV-A71 infection in neonatal mice [21]. Subsequently, Foo et al. (2007) tested the neutralizing capabilities of the antisera elicited by ninety-five diphtheria toxoid-conjugated synthetic peptides encompassing the entire VP1 region [22]. Two of the synthetic peptides designated SP55 spanning amino acids 163–177 of VP1 and SP70 spanning amino acids 208–222 of VP1 elicited high titres of neutralizing antibodies belonging to the IgG1 subtype against EV-A71. The SP55 epitope was a weaker immunogen, as the specific IgG response in mice was relatively low. Studies carried out by Xu et al. (2015) and Liu et al. (2011) confirmed SP70 as a neutralizing linear epitope of EV-A71 [23,24]. In another study, sixty-three synthetic peptides were synthesized and used to map the EV-A71 linear B-cell epitopes [25]. This study revealed the synthetic peptide designated PEP27 spanning amino acids 142–156 of VP1 carried the EV-A71 IgM-specific immune-dominant epitope. In addition, another epitope denoted as PEP23 spanning amino acids 41–55 of VP1 was identified as the EV-A71 IgG cross-reactive immune-dominant epitope. Gao et al. (2012) also identified the same region spanning amino acids 43–54 of VP1 as the IgG epitope [26]. In addition, they identified the region spanning amino acids 40–51 of VP1 as the EV-A71 IgM epitope [26].

VP2 is another capsid protein consisting of 254 amino acids. Epitope mapping studies revealed that VP2 has one neutralization epitope located in the region spanning amino acids 134–155. Liu et al. (2011) reported the region corresponding to amino acids 136–150 of VP2 (designated as VP2-28 or SP28) was able to bind an EV-A71 cross-neutralizing antibody MAB979 [23]. Subsequently, Kiener et al. (2012) reported that the region spanning amino acids 141–155 of VP2 is a cross-neutralizing epitope [27]. They identified a single amino acid mutation within the region (S144T) that led to the loss of VP2 antigenicity against EV-A71 strain C4. The same region (amino acids 141-155 of VP2) was also identified by Xu et al. (2014) to be a cross-neutralizing epitope [28].

EV-A71 VP3 protein comprises 245 amino acids. Unlike other viral structural proteins, no linear neutralizing epitope has been mapped in VP3. There is a conformational epitope within the VP3 protein known as the “Knob” region, which was identified during neutralizing monoclonal antibody screening in mice [29]. The conformational epitope spanning amino acids 55 to 69 is highly conserved among EV-A71 sub-genotypes and was recognized by monoclonal antibody 10D3. In addition, another study that immunized mice with EV-A71 sub genotype B4 mapped a conformational epitope to the highly conserved amino acid 74 of VP3, whereby this region was found to be recognized by 5H7, a therapeutic IgG antibody [30].

VP4 protein consists of 69 amino acids. VP4 adopts an extended conformation and is localized inside the virion. It is the most conserved viral structural protein. Hence, there were some studies that focussed on mapping neutralization epitopes within the VP4 protein to produce a universal vaccine against all EV-A71 sub-genotypes. Zhao et al. (2013) identified the N-terminal residues 1–20 of EV-A71 VP4 designated as VP4N20 to be a neutralizing epitope [31]. In their study, they fused VP4N20 to hepatitis B core antigen. The chimeric fusion protein was expressed in *E. coli* and was found to spontaneously assemble into chimeric virus-like-particles (VLPs). Immunization with the chimeric VLPs presenting VP4N20 was able to elicit anti-VP4N20 antibody response. The anti-chimeric VLP sera were able to confer protection against EV-A71 challenge in neonatal mice.

### 3.2. CV-A16 Epitope Mapping

Unlike EV-A71, much less is known about neutralizing linear epitopes in CV-A16. Similar to EV-A71, most of the experimentally proven neutralizing linear epitopes were mapped to the VP1 protein. Chong et al. (2012) performed the first study to map linear neutralizing epitopes in CV-A16 viral particles [32]. They identified a single linear immune-dominant epitope localized in the VP3 protein of CV-A16, spanning amino acids 176–190. However, the epitope was weakly recognized by mouse anti-sera raised against formalin-inactivated CV-A16 [32]. The failure to map more linear neutralizing epitopes in this study could be the result of the formalin treatment that changed the native surface exposed epitopes of formalin-inactivated CV-A16. The altered surface could lead to either (1) the formalin-inactivated CV-A16 losing its immunogenicity, and thus the failure to elicit antibodies, or (2) the antibodies elicited against the formalin-inactivated CV-A16 altered surface could not recognize the native amino acids of the peptides that were screened. Indeed, an earlier study performed by Liu et al. (2012) reported immunization with CV-A16 VLPs consisting of VP0, VP1, and VP3 potently elicited CV-A16 specific antibody responses. The anti-VLP sera were able to confer protection against lethal CV-A16 challenge in mice, suggesting the presence of neutralizing epitopes in the structural viral proteins [33]. Subsequently, using a set of 95 synthetic peptides encompassing the entire VP1 for reactivity with neutralizing murine antisera raised against CV-A16 VLPs, they identified six linear neutralizing epitopes localized in the VP1 protein of CV-A16 [34]. The six peptides were PEP32 (amino acids 94–108), PEP37 (amino acids 109–123), PEP55 (amino acids 163–177), PEP63 (amino acids 187–201), PEP71 that corresponded to SP70 in EV-A71 (amino acids 211–225), and PEP91 (amino acids 271–285) [34]. Other than VP1 and VP3, a neutralizing epitope was experimentally mapped to the N-terminal region of VP4 corresponding to the VP4N20 peptide in EV-A71 [35].

### 3.3. CV-A6 and CV-A10 Epitope Mapping

Apart from EV-A71 and CV-A16, other coxsackieviruses, such as CV-A6 and CV-A10, have been reported as emerging viruses to cause large scale outbreaks of HFMD [3,4]. Since these viruses are the emerging causative agents for HFMD, studies related to these viruses were reported more recently [36,37,38,39]. In one of the studies, Xu et al. (2017) determined the near-atomic resolution structure of CV-A6 A-particle in complex with a neutralizing antibody [38]. Detailed inspection of the structure mapped an immune-dominant neutralizing epitope to the surface loops of VP1, including the BC, DE, EF, and HI loops. Besides that, another study provided further insight into the structural basis of CV-A10 in complex with a neutralizing antibody designated 2G8 [36]. The atomic complex structure revealed that the Fab of 2G8 binds to a border region across three capsid proteins, VP1, VP2, and VP3. Unlike EV-A71 and CV-A16, no linear epitope mapping experiments were performed for CV-A6 and CV-A10. Due to their overall similarities, it is possible that the amino acids in CV-A6 and CV-A10 corresponded to the linear neutralizing epitopes identified in EV-A71 and CV-A16.

## 4. Antigenic Peptide-Based Vaccine

Linear peptides are attractive vaccine candidates as they may induce highly targeted immune responses, albeit its usage in isolation are often weakly immunogenic [40]. Due to their relatively small size, peptide vaccine manufacturing is relatively safe and cost effective in comparison to conventional vaccines, such as the live-attenuated and the whole-inactivated vaccines [40]. In the case of HFMD, many studies were performed to develop linear neutralizing peptide-based vaccines against the main etiological agent of HFMD, such as EV-A71 and CV-A16. Although EV-A71 formaldehyde-inactivated vaccines were recently licensed in China, the inability of these inactivated vaccines to confer protection against other HFMD causative agents, such as CV-A16, CV-A6, and CV-A10, necessitates the exploration of other vaccine platforms, such as multi-epitope peptide vaccines.

Another advantage of peptide-based vaccine is the possibility of diminishing the antibody-dependent enhancement (ADE) of viral infection. ADE refers to sub-neutralizing or non-neutralizing antibodies produced from previous primary infections or through maternal-foetal transfer that could worsen subsequent infections leading to enhanced disease severity. During secondary viral infection, the binding of pre-existing non-neutralizing antibodies to the virus enhances the entry of the virus; through the interaction between an antibody bound virus with Fc receptors (FcR) of myeloid cells, such as natural killer cells, macrophages, and B lymphocytes [41]. In the context of EV-A71 infections, ADE has been observed in both clinical and experimental settings [42,43,44]. In a study describing human IgG subclass response to EV-A71 infections, Cao et al. (2013) found the neutralizing activity of human intravenous immunoglobulin is mainly mediated by the IgG1 subclass [45]. Besides that, they found that IgG2 subclass also displayed neutralizing activity, albeit to a lesser extent as compared to IgG1. In contrast, IgG3 was found not to have neutralizing activity. Furthermore, they demonstrated that the IgG3 subclass enhanced EV-A71 infection [45]. These findings are critical for the rational design of peptide-based vaccines that only induce high levels of the neutralizing antibody responses (e.g., IgG1 and IgG2), but not the non-neutralizing antibody responses (e.g., IgG3), towards EV-A71 or other HFMD etiological agents.

### 4.1. Peptide-Based Vaccines Against EV-A71

Several studies reported the efficacies of linear antigenic peptide-based vaccines against EV-A71. For instance, a diphtheria toxoid-conjugated synthetic peptide comprising the EV-A71 SP70 neutralizing epitope was able to confer 80% passive protection in mice upon lethal challenge and elicited cross protective neutralizing antibodies (1:32) against EV-A71 sub-genotypes B2, B5, C2, and C4 [46]. Subsequently, Tian et al. (2012) reported the incorporation of the EV-A71 SP70 epitope within surface-exposed domains of the adenovirus type 3 hexon [47]. The construct was able to elicit humoral responses specific to the SP70 epitope. They also found that antisera raised against the construct were able to confer 70% protection of mice upon lethal EV-A71 challenge. Even though the protection was less than the diphtheria toxoid-conjugated EV-A71 SP70, this study demonstrated the possibility of using adenovirus to display the EV-A71 neutralizing epitope.

Another study focussed on the design of a tandem multi-linear neutralizing peptide vaccine. Li et al. (2014) designed a novel recombinant tandem multi-linear neutralizing epitope of EV-A71 designated as mTLNE [48]. The mTLNE comprised three linear neutralizing epitopes of SP55, SP70, and SP28, which were sequentially connected by a (Gly_4_Ser)_3_ linker. The mTLNE also contained thioredoxin (Trx) at the N-terminal to improve solubility and immunogenicity. Additionally, six histidine (His) were added at the C-terminal for affinity chromatography purification of the *E. coli* expressed proteins. Following immunization in adult mice, the mTLNE elicited EV-A71 specific IgG antibodies. Passive transfer of the anti-mTLNE sera conferred 100% protection against lethal EV-A71 challenge in mice. This is a marked improvement in comparison to the 80% protection conferred by the anti-diphtheria toxoid-conjugated EV-A71 SP70 sera and 70% protection conferred by antisera raised against recombinant adenovirus carrying the SP70 epitope in previous lethal challenge studies [46,47]. Although the neutralizing titres elicited by SP55, SP70, and the mTLNE peptide vaccine were relatively low in comparison to EV-A71 VLPs and the formaldehyde-inactivated EV-A71, the advantage of this approach should be examined. For instance, immunization studies in mice revealed that SP55, SP70, and the mTLNE peptide vaccine induced robust IgG1 response and not IgG3 [22,48]. This indicated these peptides would not contribute to the ADE phenomenon if incorporated into future peptide-based vaccine development.

### 4.2. EV-A71/CV-A16 Peptide-Based Bivalent Vaccines

Consistent with the low amino acid sequence identity of 20.7% between EV-A71 and CV-A16 capsid proteins (VP1, VP2, VP3, and VP4), vaccine candidates that were able to confer protection against EV-A71 were weakly cross-protective against CV-A16 infection [49,50]. For instance, VLPs and inactivated EV-A71 or CV-A16 monovalent vaccines failed to provide sufficient cross protection against their enteroviral counterparts [50]. To address this issue, an EV-A71/CV-A16 bivalent vaccine was produced by mixing equivalent doses of EV-A71 and CV-A16 VLPs or the inactivated EV-A71 and CV-A16 viruses. The EV-A71/CV-A16 bivalent vaccine was able to elicit cross-neutralizing antibodies and the immune sera from vaccinated animals could confer passive protection against EV-A71 and CV-A16 lethal challenges. However, the large-scale growth and production of either of the viruses or the VLPs constituting the bivalent vaccine will not be cost effective. Moreover, chemical treatments have been shown to alter the antigenic surface of CV-A16 [32,51]. Considering these problems, it is better to produce a multi-epitope antigen or a chimeric antigen that can elicit neutralizing antibodies against both EV-A71 and CV-A16.

Based on the 80% sequence identity between SP70 and PEP71 epitopes, coupled with structural studies that revealed these epitopes were displayed on the virion surface, Zhao et al. (2015) designed a peptide-based EV-A71/CV-A16 bivalent vaccine [52,53,54,55]. They constructed a chimeric EV-A71 VLP designated as ChiEV-A71 VLP, in which the SP70 of EV-A71 was replaced with PEP71 of CV-A16 [52]. The yeast expressed ChiEV-A71 VLPs were found to induce robust Th1/Th2-dependent immune responses against EV-A71 and CV-A16 in mice. Besides that, passive immunization with anti-ChiEV-A71 VLP sera conferred full protection against lethal EV-A71 and CV-A16 challenges. Subsequent studies provided insights into the structural basis of EV-A71 and CV-A16 neutralizations by ChiEV-A71 VLPs [56]. The substitution of the SP70 segment in EV-A71 with the corresponding PEP71 region from CV-A16 led to surface charge potential of the modified region to shift from negative to neutral. The surface charge potential shifts coupled with the amino acid changes (K215L, E217A, K218N, E221D) most likely accounted for ChiEV-A71 VLP dual neutralization capabilities of EV-A71 and CV-A16. Importantly, this is the first study utilizing an antigenic peptide to produce a bivalent vaccine in the form of a chimeric VLP.

In another effort to produce a bivalent vaccine, Xu et al. (2015) used hepatitis B virus core protein to construct a bivalent chimeric VLP presenting the SP70 epitope and VP2 epitope spanning amino acids 141 to 155 of EV-A71 [24]. They discovered that the chimeric VLP designated as HBc-E1/2 elicited high IgG and neutralization titres against both EV-A71 and CV-A16. Furthermore, passive immunization with HBc-E1/2 protected neonatal mice against lethal EV-A71 and CV-A16 challenges. The VP2 epitope (amino acids 141 to 155) was shown to be immune-dominant in both EV-A71 and CV-A16. They found that the anti-VP2 epitope elicited antibodies that could bind to both EV-A71 and CV-A16 viral particles, whereas antibodies from the anti-SP70 epitope did not bind to CV-A16 viral particles. These studies highlighted the possibility of employing linear epitopes identified from both EV-A71 and CV-A16 to produce a bivalent vaccine against both viruses.

### 4.3. Peptide Insertion Sites in EV-A71 VLP

To date, VLP is one the most suitable and widely used platform technologies to display antigenic peptides for vaccine development [57]. Advantages of VLP as a vaccine platform include: (1) its 20–100 nm diameter size which allows easy entry into the lymphatic vessels, passive drainage to the subcapsular lymph nodes, and optimal uptake by antigen presenting cells; and (2) the geometry that contributes to their ability to activate B-cells as the epitopes presented on the repetitive and rigid structures of VLPs could extensively interact with antigen presenting cells, leading to stimulation of B-cells and elicitation of robust antibody responses. Besides directly substituting the corresponding regions in a VLP with another region from a different virus to form a bivalent vaccine, regions in the VLP that were exposed or externalized on the viral capsid could be exploited to display B-cell epitopes from other enteroviruses to form multivalent vaccines. A recent study has reported repeated administration of an EV-A71 VLP vaccine which were well tolerated in rabbits and immunogenic. This further supported the use of EV-A71 VLP as a vaccine platform to display antigenic peptides [58].

In EV-A71 research, Arita et al. (2007) firstly identified that a histidine tag could be inserted into the BC-loop of VP1 between amino acids 100 and 101 without affecting the infectivity of the engineered EV-A71 [59]. Subsequently, Lyu et al. (2013) inserted nucleating peptides at the same position to produce a thermostable attenuated EV-A71 vaccine [60]. The insertion of peptides endowed the virus with the capacity to generate an exterior calcium phosphate shell. Intriguingly, the modified surface enabled the virus to exhibit improved thermostability and immunogenicity. They also determined the structures of two naturally occurring empty EV-A71 particles, one from a clinical C4 strain and the other from the recombinant virus containing a foreign peptide in the BC loop of VP1 [55]. Detailed inspection of the structures revealed the inserted foreign peptide was well displayed on the viral particle surface without significant structural perturbations of the viral capsid. In addition, comparison of the uncoating intermediates of EV-A71 and recombinant EV-A71 virus demonstrated that the foreign peptide insertion did not affect the virus uncoating process. Furthermore, the crystal structure of yeast-produced EV-A71 revealed that the VLP produced using the yeast expression system (PDB code: 4YVS) shared very high structural similarity with the empty particles (PDB code: 4RQP). Superimposition using PyMOL of the VP1, VP3, and VP0 onto the corresponding viral structural proteins on EV-A71 empty virus particle resulted in a calculated root mean square deviation (r.m.s.d) of 0.504 Å over 560 Cα atoms. Based on the structural and functional evidence, the BC-loop of EV-A71 was identified to be a suitable peptide insertion site to develop multivalent vaccines.

In addition to the BC-loop, other insertion sites should be identified to produce trivalent or tetravalent HFMD vaccines against other HFMD causative agents. Based on the available EV-A71 structures in the protein database, the EF loop of VP0 (or VP2) corresponding to amino acids 202 to 214 might be another suitable site for foreign peptide insertion (Figure 1). This region is located at the surface of the viral capsid. As demonstrated previously by Lyu et al. (2015), the surface-exposed loop was an ideal site for foreign linear peptide insertion [55]. However, peptide insertions might affect the ability of recombinant protein to assemble into VLPs. Thus, structural and functional validations using techniques such as X-ray crystallography and Cryo-electron microscopy are crucial to ensure that the peptide insertions do not perturb the overall structure of the VLP.

### 4.4. Tetravalent Peptide-Based HFMD Vaccine

Previous attempts were made to use the traditional VLP-based platform to produce a tetravalent HFMD vaccine. For instance, Zhang et al. (2018) produced each single VLP of EV-A71, CV-A16, CV-A6, and CV-A10 using a baculovirus-insect cell expression system, and then combined them to formulate a tetravalent HFMD vaccine [61]. They found the tetravalent vaccine elicited antigen specific and long lasting antibody responses in addition to passive protection conferred by the tetravalent vaccine-immunized sera against single or mixed infections with EV-A71, CV-A16, CV-A6, and CV-A10 in mice. However, large scale productions of the four VLPs will be costly and time-inefficient in comparison to the production of a single VLP carrying peptide epitopes from the four different viruses.

Similar to EV-A71 and CV-A16, other Enterovirus A serotypes were found to carry the SP70 motif. Based on sequence analysis of several Enterovirus A serotypes, the SP70 motif can be denoted as YPTFGX_1_HPX_2_X_3_X_4_X_5_X_6_X_7_Y (Figure 2). The motif consists of the highly conserved amino acids Tyr, Pro, Thr, Phe, Gly (YPTFG), followed by X_1_ (which can either be polar uncharged amino acids or negatively charged amino acids), the highly conserved His and Pro, followed by X_2_, X_3_, X_4_, X_5_, X_6_, X_7_, and ending with a highly conserved Tyr. Amino acids X_2_ and X_3_ can be any amino acid, X_4_ and X_7_ can be any of the polar uncharged or charged amino acids, X_5_ can either be an Asp or an Asn, while X_6_ can be any hydrophobic amino acid (Leu, Val, Phe, Ile).

Based on previous findings of the bivalent EV-A71/CV-A16 vaccine, the SP70 linear epitope from EV-A71, CV-A16, CV-A6, and CV-A10 could be incorporated into a VLP to produce a tetravalent HFMD vaccine [52]. Inspection of the viral particle structures of EV-A71, CV-A16, CV-A6, and CV-A10 revealed all viral particles to be structurally very similar. Crucially, the SP70 motif was present within the GH loop of each of the viral particles, hence, it is possible to insert the SP70 motif of each virus into a single VLP. For example, the EV-A71 VLP is a suitable VLP to be used to construct the HFMD tetravalent vaccine, as it is the most widely studied amongst all the enteroviral VLPs. In addition to the substitution of the SP70 motif of EV-A71 with the corresponding CV-A16 epitope in the EV-A71 VLP, the SP70 motif from CV-A6 could also be inserted into the BC-loop insertion site and another SP70 motif from CV-A10 could be inserted into the site in the EF loop of VP0 or VP2. The insertion site should be experimentally and structurally validated to ensure that the SP70 peptide insertion does not disrupt the global structure of the EV-A71 VLP. Furthermore, studies should be performed to confirm that the SP70 motifs of both CV-A6 and CV-A10 are able to elicit neutralization antibodies, similar to the SP70 motif present in either EV-A71 or CV-A16.

Another approach that should be considered is to expand the tandem multi-linear neutralizing epitope (mTLNE) vaccine strategy to produce a multivalent vaccine. It is possible to design a peptide vaccine containing multiple SP70 epitopes from EV-A71, CV-A16, CV-A6, and CV-A10 that are linked together to produce a tetravalent HFMD peptide-based vaccine that is protective against the four viruses. Besides that, the mTLNE vaccine approach could be presented on nanoparticles or on VLPs which might improve the immunogenicity and efficacy of the tetravalent vaccine.

Besides EV-A71 VLP, hepatitis B core antigen virus-like-particle (HBcAg-VLP) is an attractive alternative VLP to display multiple epitopes of HFMD etiological agents. HBcAg-VLP is one of the best characterized VLPs. It possesses three possible peptide or antigen insertion sites, which are located at the N terminus, the C terminus, and the immune-dominant loop region (between Asp79 and Pro79) [62]. For the N and C termini insertions, a maximum of 100 amino acids can be inserted into each insertion site [62]. Although the inserted antigens at the C and N termini will be immunogenic, HBcAg-VLP will retain its strong intrinsic immunogenicity as immune responses would target the immune-dominant loop region. Therefore, to achieve the highest immunogenicity, the insertion at the immune-dominant loop region is most favourable. Insertion of an antigen within the immune-dominant loop region will abrogate HBcAg-VLP intrinsic immunogenicity and directs immune responses towards the inserted foreign antigen. Besides that, the antigen insertion capacity of this region is very high, as demonstrated by the successful insertion of the green fluorescent protein (238 aa) and the dimeric outer surface lipoprotein C of *Borrelia burgdorferi* (~210 aa) [63,64]. 

Crucially, Wu et al. (2017) demonstrated that HBcAg-VLP inserted with two linear epitopes from EV-A71 (SP70 epitope and amino acids 141-155 of VP2) and one epitope from glycoprotein E (amino acids 121–135) of varicella-zoster virus (VCV) in a tandem manner was able to self-assemble into a VLP [65]. The VLP designated as HBc-V/1/2 elicited a balanced antibody response towards the three epitopes. Importantly, the HBc-V/1/2 antisera could neutralize EV-A71, CV-A16, and VCV. Moreover, the HBc-V/1/2 antisera were able to protect neonatal mice against lethal challenges of EV-A71 and CV-A16. Based on these studies, the combination of the HBcAg-VLP platform with the mTLNE strategy might yield an effective tetravalent HFMD vaccine against EV-A71, CV-A16, CV-A6, and CV-A10. For instance, a highly immunogenic linear epitope, such as the SP70 epitope from each of the four HFMD etiological agents, could be linked together to form a mTLNE. The mTLNE can be inserted into the immune-dominant region of HBcAg-VLP to develop a multivalent HFMD vaccine.

Theoretically, there are many advantages of using HBcAg-VLP than EV-A71 VLP to produce a multivalent HFMD vaccine. Firstly, the use of HBcAg-VLP is more cost-effective, as HBcAg-VLP could be produced in a bacterial expression system such as *E. coli*. This system will incur lower production cost than yeast, insect, or mammalian cell expression systems that are required to produce the EV-A71 VLP. Secondly, HBcAg-VLP has a higher peptide insertion capacity (238 aa could be inserted in its immune-dominant loop region) in comparison to EV-A71 VLP that could only receive a peptide of around 20 amino acids in its BC loop region. Nonetheless, the suitability of HBcAg-VLP and EV-A71 VLP to carry multiple epitopes as an effective multivalent HFMD vaccine should be further tested.

## 5. Neutralizing Antibodies Against HFMD Etiological Agents

### 5.1. EV-A71 Neutralizing Antibodies

Neutralizing antibodies play major roles in immune-protection against viral infections. In EV-A71 infection, robust induction of EV-A71-specific IgM and IgG were detected during the first week of infection and the responses peaked four to seven days after the onset of symptoms [66]. Passive transfer of neutralizing anti-sera was reported to protect mice against lethal EV-A71 challenge, indicating the importance of neutralizing antibodies in anti-EV-A71 immunity [46]. Importantly, comprehensive understanding of the molecular basis of EV-A71 neutralization by antibodies would aid in the rational design of an effective vaccine against EV-A71. In order to understand neutralizing antibodies elicited by EV-A71, many groups generated neutralizing murine monoclonal antibodies (mAbs) using immunogens, such as inactivated EV-A71, live EV-A71, EV-A71 VLPs, or the linear neutralizing epitopes derived from EV-A71 (Table 2).

Many of the EV-A71 neutralizing antibodies were generated by immunizing mice with live EV-A71. For instance, Chang et al. (2011) inoculated mice with the live EV-A71/E59 genotype B4 strain to generate mAbs [67]. The study identified four mAbs belonging to the IgG2a subclass designated as N1, N3, N4, and N6. From epitope mapping studies, the four mAbs were found to recognize amino acids 211 to 225 of the VP1 protein corresponding to the SP70 epitope. The mAbs were able to fully cross-neutralize EV-A71 sub-genotypes B4 and B5 in the rhabdomyosarcoma (RD) cells and it was noted that these mAbs could work synergistically, as complete inhibition was achieved at a markedly lower concentration of the pooled mAbs in comparison to a single mAb in RD cells. In addition, Xu et al. (2014) generated a mAb designated as BB1A5 belonging to the IgG2a isotype by immunizing mice with the live EV-A71 strain 52-3 [28]. BB1A5 recognized a linear epitope spanning amino acids 136–155 in the VP2, and passive transfer of BB1A5 was able to provide 100% protection for the new-born mice challenged with a mouse-adapted EV-A71 strain. Besides that, another group identified a mAb designated as 3D1 belonging to the IgM isotype generated by immunizing mice with a mouse-adapted EV-A71 strain [68]. They demonstrated that passive immunization using 3D1 prevented central nervous system (CNS) infection. However, the exact binding site of 3D1 was unknown.

Additionally, inactivated EV-A71 was utilized to generate mAbs against EV-A71. For instance, Chang et al. (2010) identified mAb 4E8, which was generated by immunizing mice with an inactivated EV-A71 Henan2 (Hn2) genotype C4 strain [69]. Using synthetic peptide mapping, they identified mAb 4E8 to recognize peptides containing amino acids 240–250 and 250–260 of VP1 by ELISA. In the in vivo protection study, mAb 4E8 was able to provide partial protection against EV-A71 lethal challenge in mice. In another study, Kiener et al. (2014) immunized mice with a binary ethyleneimine (BEI)-inactivated EV-A71-B4 strain to generate mAb 10D3 belonging to the IgM isotype [29]. Dot blot assay coupled with mutational analysis revealed that mAb 10D3 recognized a conformational epitope at the highly conserved knob region of VP3. MAb 10D3 was able to confer in vivo passive protection against EV-A71 infection. Moreover, neutralizing mAb51 belonging to the IgM isotype was generated by immunizing mice with binary BEI-inactivated EV-A71 [70]. The mAb51 was able to provide 100% in vivo passive protection in lethal challenge with EV-A71 in mice. Epitope mapping revealed that the mAb51 targeted amino acids 215–219 in VP1, which is part of the SP70 epitope. Furthermore, Deng et al. (2015) generated IgM isotype mAb designated as 2G8 by immunizing mice with heat-inactivated EV-A71 [71]. The mAb was able to neutralize EV-A71 infection at the attachment stage. Biochemical assays mapped the 2G8 epitope to the SP70 epitope. They also demonstrated that the L220A mutation within the SP70 epitope completely abolished the ability of 2G8 to bind and neutralize EV-A71. Another mAb designated as MA28-7 was generated by immunizing mice with formalin-inactivated EV-A71 strain 1095 [72]. Structural studies of MA28-7 in complex with EV-A71 revealed that the mAb recognized a conformational epitope comprising several amino acids in VP1, including Gly145, Glu98, Lys242, and Lys244. D6 and A9 were two other mAbs that were generated by immunization of mice using inactivated EV-A71 [53]. Both mAbs were able to protect RD cells from EV-A71 infection. Most recently, a mAb designated as 5H7 belonging to the IgG isotype was generated by immunizing mice with inactivated EV-A71-B4 strain [30]. Epitope mapping revealed that 5H7 recognized a conformational epitope within the VP3 protein harboring the highly conserved amino acid Ser74. Interestingly, the highly conserved Ser74 is adjacent to the highly conserved knob region of VP3, which was recognized by 10D3. MAb 5H7 was able to neutralize major EV-A71 genogroups, including A, B4, C2, and C4. Chimeric 5H7 mAb was constructed by fusing the VH and VL regions of murine mAb 5H7 to human kappa and IgG1 constant regions. Passive immunization using the resulting mouse-human chimeric 5H7 protected 100% of mice challenged with EV-A71 B4 strain prophylactically and therapeutically.

Other than live and inactivated viruses, VLPs have been used to generate mAbs. For example, Lin et al. (2015) immunized mice with VLPs of EV-A71 produced in yeast [73]. From the immunization, two mAbs were isolated, designated as D4 and G12. The mAbs bound to the VP1 of EV-A71, however, the exact binding site of the mAbs was unknown. In an in vitro neutralization assay, both mAbs were able to protect RD cells from CPE upon EV-A71 infection.

Several mAbs were generated by immunizing mice with synthetic peptides. For example, Li et al. (2009) immunized mice using SP70 and SP55 peptides to isolate twelve IgG mAbs specific for SP70 and SP55 [74]. Out of the twelve mAbs, only mAb 22A12, which specifically recognized SP70 peptide, provided complete protection of RD cells from EV-A71 viral infection at a neutralizing titre of 1:16, albeit this titre was quite low in comparison to the 1:48 neutralization titre produced by antiserum from rabbit immunized with the heat-inactivated virus [74]. In another study, a phage display technique was used to produce a recombinant mAb targeting VP4N20 epitope in the VP4 protein [35]. In vitro neutralizing assays revealed that the VP4N20 mAb prevented the development of cytopathic effects in RD cells infected with either EV-A71 or CV-A16. This is consistent with the 90% amino acid identity of VP4N20 epitope in both EV-A71 and CV-A16.

### 5.2. CV-A16, CV-A6 and CV-A10 Neutralizing Antibodies

Unlike EV-A71, few neutralization mAbs have been generated and identified against CV-A16, CV-A6, and CV-A10. In a study performed by Ren et al. (2015), mAb C33 was generated by immunizing mice with inactivated CV-A16 [53]. The mAb C33 was able to provide complete protection against the development of cytopathic effects in Vero cells infected with CV-A16. For CV-A6, Yang et al. (2016) immunized mice using live CV-A6 to generate mAbs [39]. They identified mAb 1D5 that could fully protect mice from lethal CV-A6 challenge. The mAb also displayed good therapeutic effects in the mice infected with CV-A6. Structural studies using cryo-electron microscopy revealed that mAb 1D5 bound to the immune-dominant antigenic site within VP1 comprising the BC, DE, EF, and HI loops [38]. For CV-A10, a neutralizing mAb designated as 2G8 was obtained by immunizing mice with CV-A10 mature virions [36]. The mAb displayed high binding efficiency and potent neutralization activity against CV-A10 infection based on ELISA and in vitro neutralization in RD cells. In vivo experiments in the murine model revealed that 2G8 exhibited strong prophylactic and therapeutic properties, whereby all mice administered with 2G8 before or after the CV-A10 lethal challenge survived. Structural characterization using cryo-electron microscopy revealed that 2G8 recognized the VP1 C-terminus, VP2 EF loop, and VP3 AB loop.

### 5.3. Neutralizing Antibodies as Potential Prophylactic or Therapeutic Agents against Enteroviruses Causing HFMD

Antibodies can be administered as a therapeutic or prophylactic agent. Monoclonal antibody therapy is crucial for immunocompromised individuals, as vaccinations are only suitable for individuals with an immune system capable of eliciting responses against the administered vaccine [75]. The antibodies to be used in mAb therapy should have high neutralizing efficacy coupled with broad neutralizing spectrum against all or most of the viral subgenotypes. Ideally, the antibody therapy should be able to neutralize multiple viruses causing the disease for it to be effective against multiple etiological agents of HFMD.

As shown in Table 2, only four mAbs (3D1, 10D3, mAb51, and 5H7) were experimentally proven to display broad neutralizing spectrum. Interestingly, three of the four mAbs, with the exception of 5H7, belonged to the IgM isotype. This observation is consistent with other studies and observations made by several research groups that suggested antibody neutralizing capabilities were dependent on their isotypes [71,75]. For example, Lim et al. (2012) reported that in contrast to the IgM isotype that possessed neutralizing activity in vitro, IgG1 did not have neutralizing activity [70]. Besides that, Liu et al. (2011) reported that IgG2a mAb displayed only modest neutralization activity against EV-A71 [23]. Thus, it can be speculated that neutralizing capability of these mAbs is isotype-dependent. In addition, three of the universally neutralizing mAbs, 10D3, 5H7, and mAb51, recognized a region or a sequence that was highly conserved in all 11 subgenotypes, which was the knob region in VP3, the highly conserved Ser74 adjacent to the knob region in VP3, and a short sequence in the SP70 epitope (KQEKD), respectively [29,30,70]. This most likely contributes to their universal neutralizing capability. Based on the evidence, it is possible that other mAbs, such as mAb VP4N20, which recognized an epitope that is highly conserved in EV-A71 subgenotypes and in the CV-A16, might have a broad neutralizing spectrum. However, neutralization by the mAb was only tested against limited EV-A71 subgenotypes. Therefore, it is imperative to further characterize the neutralizing spectrum of all the mAbs identified so far to establish a comprehensive understanding of neutralizing mAbs against the etiological agents of HFMD.

A major issue regarding the potential of using mAb clinically is that most of the mAbs were derived from mice. Such mouse-derived mAbs would lead to immune rejection in humans, as they might be recognized as foreign antigens by the human immune system [76]. Besides that, murine mAbs were unable to induce downstream human immune responses due to the inability of human immune cells, such as macrophages, to recognize the Fc region of murine mAbs. To circumvent this problem, the humanization of murine mAbs is mandatory prior to passive immunization using mouse-derived mAbs in humans. Taking dengue virus mAb513 as an example, which is currently undergoing clinical trials in Singapore, few amino acid changes were made, including Thr33Val mutation and Ser26 deletion, which resulted in the murine mAb 4E11 to be humanized [77]. In another example, complementary determining region (CDR) grafting and variable (V) region replacement were performed to convert the influenza virus subtype H5N1 antibody m8A8 to humanized 8A8 with minimal murine immunogenicity [78]. Based on this evidence, it is possible to humanize murine mAbs of enteroviruses for future medical applications (Table 2).

The design of an effective prophylactic mAb to prevent HFMD needs to account for the multiple etiological agents that could be involved in HFMD outbreaks. The multi-domain antibody approach could be an attractive approach to target several etiological agents of HFMD. Recently, there has been one research group that designed a multi-domain antibody against multiple strains of influenza virus. Laursen et al. (2018) linked multiple antibodies to design a prophylactic antibody against influenza A and B strains designated as MD3606 [79]. MD3606 was able to confer protection in mice against influenza A and influenza B infections. Thus, it may be possible to link four mAbs that were able to neutralize each of the HFMD etiological agents (EV-A71, CV-A16, CV-A6, and CV-A10) to design a tetravalent HFMD prophylactic antibody.

## 6. Conclusions

The global burden of HFMD is increasing and the trend will continue until an effective vaccine against multiple HFMD causative agents is developed. Although there are three formaldehyde-inactivated EV-A71 vaccines licensed by the China FDA, the inactivated monovalent vaccines are only protective against EV-A71 and not the other HFMD causative agents. Apart from developing a bivalent EV-A71 and CV-A16 vaccine, research efforts are now focussed on developing a multivalent HFMD vaccine. Comprehensive evaluations of humoral responses elicited by epitopes from EV-A71, CV-A16, CV-A6, and CV-A10 are mandatory for a successful multivalent HFMD vaccine development. The multivalent vaccine should provide protection against significant HFMD Enterovirus A serotypes in a balanced manner and should ideally avoid the ADE effect. Future studies to utilize other vaccine platforms, such as live-attenuated, recombinant protein subunit, viral vectored, and peptide-based vaccines are needed, and will definitely open up new avenues to develop fully efficacious and safe vaccines against HFMD. The multi epitope peptide-based HFMD vaccines are gaining attention due to their safety, easy large-scale production, and low cost. With the rising concern about the current HFMD outbreaks in Asia, the future will see more development of the HFMD tetravalent vaccines, especially on a peptide-based multivalent HFMD vaccine platform, if low immunogenicity of such vaccines could be overcome. Finally, further identification and generation of neutralizing antibodies against etiological agents of HFMD are warranted, as prophylactic and therapeutic antibodies against viruses are gaining popularity, as demonstrated in the development of mAb therapies against dengue and influenza viruses.

## Figures and Tables

**Figure 1 ijms-20-01256-f001:**
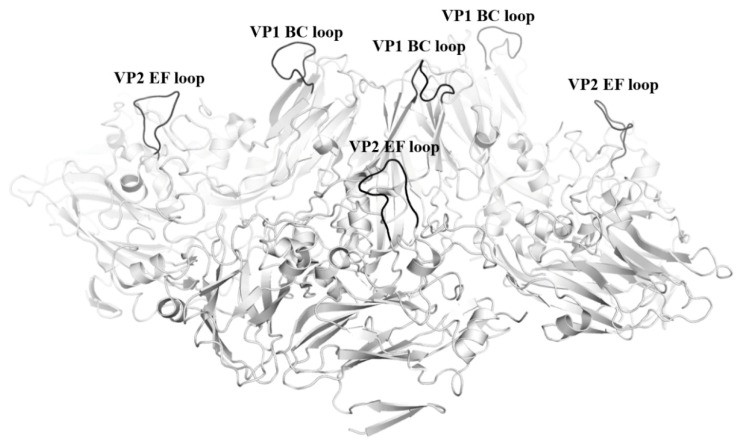
Position of VP1 BC loop and VP2 EF loop on the EV-A71 viral capsid. Representation of the EV-A71 VLP viral capsid. VP1, VP2 and VP3 structural proteins are shown in grey. The VP1 BC loop and VP2 EF loop are shown in black and labelled. PDB code: 4RQP.

**Figure 2 ijms-20-01256-f002:**
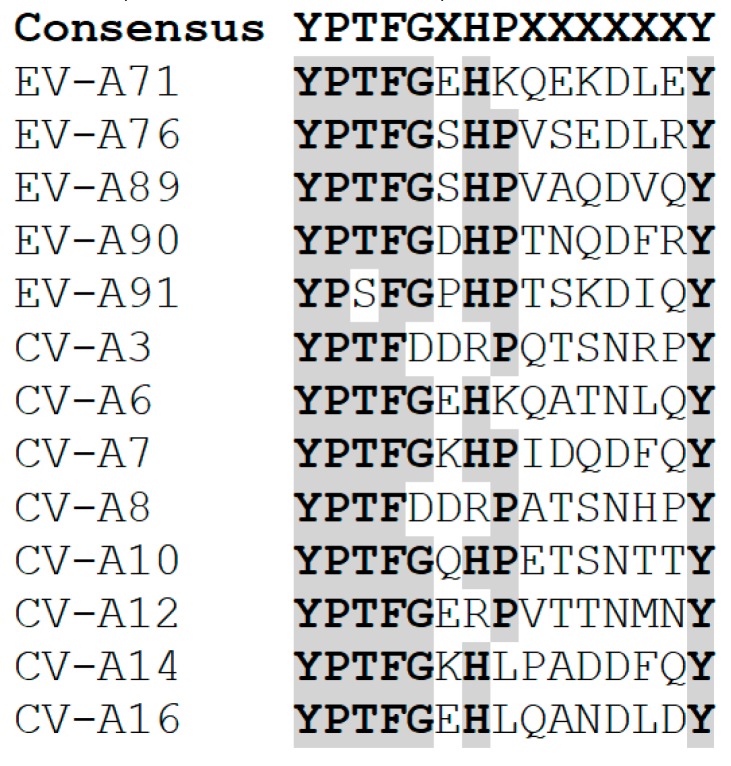
Multiple sequence alignment of SP70 motif from various Enterovirus A viruses. The SP70 motif of Enterovirus A species can be denoted as YPTFGX_1_HPX_2_X_3_X_4_X_5_X_6_X_7_Y. The consensus amino acid is highlighted in grey and in bold.

**Table 1 ijms-20-01256-t001:** Experimentally mapped linear epitopes in the structural proteins of EV-A71 and CV-A16.

Virus	Region	Name and Position	Peptide Sequences	Epitopes	Ref.
EV-A71	VP1	PEP23 (aa 41–55)	TGEVPALQAAEIGAS	IgG	[25]
		VP1–15 (aa 43–54)	KVPALQAAEIGA	IgG	[26]
		VP1–14 (aa 40–51)	DTGEVPALQAAE	IgM	[26]
	VP1	PEP27 (aa 142–156)	PTGEVVPQLLQYMFV	IgM	[25]
	VP1	SP55 (aa 163–177)	PESRESLAWQTATNP	IgG	[22]
	VP1	SP70 (aa 208–222)	YPTFGEHKQEKDLEYC	IgM and IgG	[22]
		VP1 (aa 208–222)	YPTFGEHKQEKDLEYC	IgG	[24]
		VP1–43 (aa 211–220)	FGEHKQEKDL	IgG	[23]
	VP2	VP2–28 (aa 136–150)	AGGTGTEDSHPPYKQ	IgG	[23]
		7C7 (aa 142–146)	EDSHP	IgG	[28]
		VP2 (aa 141–155)	TEDSHPPYKQTQPGA	IgG	[27]
	VP4	VP4N20 (aa 1–20)	GSQVSTQRSGSHENSNSATE	Neutralizing	[31]
CV-A16	VP1	PEP32 (aa 94–108)	TMPTMGTQNTDGYAN	IgG	[34]
		PEP37 (aa 109–123)	WDIDLMGYAQLRRKC	IgG	[34]
		PEP55 (aa 163–177)	PTSRDSFAWQTATNP	IgG	[34]
		PEP63 (aa 187–201)	PAQVSVPFMSPASAY	IgG	[34]
		PEP71 (aa 211–225)	FGEHLQANDLDYGQC	IgG	[34]
		PEP91 (aa 271–285)	YLFKTNPNYKGNDIK	IgG	[34]
	VP3	VP3–41 (aa 176–190)	HYRAHARAGYFDYYT	IgG	[32]
	VP4	VP4N20 (aa 1–20)	GSQVSTQRSGSHENSNSASE	Neutralizing	[35]

aa = amino acid.

**Table 2 ijms-20-01256-t002:** Neutralizing monoclonal antibodies against EV-A71, CV-A16, CV-A6, and CV-A10.

Virus	Name	Epitope Type	Epitope/Region Recognised	Isotype	Ref.
EV-A71	N1	Linear	Recognized a section of the SP70 epitope. Neutralized 2 subgenotypes tested (B4 and B5).	IgG2a	[67]
	N3	Linear	Recognized a section of the SP70 epitope. Neutralized 2 subgenotypes tested (B4 and B5).	IgG2a	[67]
	N4	Linear	Recognized a section of the SP70 epitope. Neutralized 2 subgenotypes tested (B4 and B5).	IgG2a	[67]
	N6	Linear	Recognized a section of the SP70 epitope. Neutralized 2 subgenotypes tested (B4 and B5).	IgG2a	[67]
	BB1A5	Linear	Recognized amino acids 136–155 in VP2 (AGGTGTEDSHPPYKQTQPGA). Neutralized B3, B4, C2 and C5 subgenotypes.	IgG2a	[28]
	3D1	Unknown	Recognized VP1. Neutralized all 8 subgenotypes tested (C1, C2, C3, C4, C5, B3, B4, and B5).	IgM	[68]
	4E8	Unknown	Recognized two peptides P25 (aa2 40–250) and P26 (aa 250–260) of VP1. Neutralized C4 subgenotype (only subgenotype tested).	IgG	[69]
	10D3	Conformational	Recognized the knob region in VP3. Neutralized all 11 subgenotypes of EV-A71.	IgM	[29]
	mAb51	Linear	Recognized a portion of the SP70 epitope (KQEKD). Neutralized all 11 subgenotypes of EV-A71.	IgM	[70]
	2G8	Linear	Recognized the SP70 epitope. Neutralized C4 subgenotype.	IgM	[71]
	MA28-7	Conformational	Recognized the region comprising Gly145, Glu98, Lys242 and Lys244 in VP1. Neutralized 6 genotypes tested (A, B1, B3, B4, and C2).	N/A	[72]
	D6	Unknown	Neutralized subgenotype C4 (only subgenotype tested).	N/A	[53]
	A9	Unknown	Neutralized subgenotype C4 (only subgenotype tested).	N/A	[53]
	5H7	Conformational	Recognized conformational epitope harbouring Ser74 in VP3. Neutralized all 11 subgenotypes tested.	IgG	[30]
	D4	Unknown	Neutralized subgenotype C4 (only subgenotype tested).	IgG	[73]
	G12	Unknown	Neutralized subgenotype C4 (only subgenotype tested).	IgG	[73]
	22A12	Linear	Neutralized subgenotype C4 (only subgenotype tested).	IgG	[74]
	VP4N20 mAb	Linear	Recognized the VP4N20 epitope in VP4. Neutralized EV-A71 tested (genogroup A and subgenotype C4) and CV-A16.	N/A	[35]
CV-A16	VP4N20 mAb	Linear	Recognized the VP4N20 epitope in VP4. Neutralized EV-A71 tested (genogroup A and subgenotype C4) and CV-A16.	N/A	[35]
	C33	Unknown	Recognized the SP70 epitope. Neutralized genotype B of CV-A16 (only one tested).	N/A	[53]
CV-A6	1D5	Conformational	Recognized a region in VP1 comprising BC, DE, EF, and HI loops.	IgG	[38,39]
CV-A10	2G8	Conformational	Recognized the VP1 C-terminus, VP2 EF loop, and VP3 AB loop.	IgG	[36]

N/A: Not reported.

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
