# Peer review of "Advances in Antigenic Peptide-Based Vaccine and Neutralizing Antibodies against Viruses Causing Hand, Foot, and Mouth Disease"

_ijms, 2019, doi:10.3390/ijms20061256_

Reviewer 1 Report

Good and comprehensive review of vaccination approaches to the prevention of virus-induced HFMD in humans (especially - but not limited to - Asia). Appropriately, the possible therapeutic use of monoclonal antiviral antibodies is also reviewed.

The MS is well organized and written, Tables are clear, references complete.

Minor points:

Line 49, ref 14: since the dose of formaldehyde for virus inactivation is so low, the concern is only theoretical and does not have relevance for human applications (see Plotkin, Vaccines 2018). This should clearly reported in a Journal devoted to Virologists and immunology experts.

Lines 55 and 57: past tense nont appropriate "discussed" etc.

Line 71: "are all", remove

Line 295: ref 62 "self-biomineralizing". The term needs clarification. The meaning is NOT obvious, nor well accepted in the literature. Try explaining (if needed).

Line 333: cost, change into costly

Line 346: speices: change into species

Line 454: inhibition of CPE by 95% - the concept is possibly derived from the ref. Its is however clear that "in vitro" an antibody is either neutralizing (ie not producing CPE) or non-neutralizing (poorly-neutralizing), ie letting the CPE start and proceed to the final death of cells (given enough time and adequate culture conditions, of course!). Thus, inhibiting is not correct. Please modify the sentence. The same applies to lines 466-467. Antibody prevents CPE or protects from infection (thus preventing CPE).

Line 485: Vero cells

Line 571: abbreviations - please put in alphabetic order.

Author Response

Response to Reviewer 1 Comments

Point 1: Line 49, ref 14: since the dose of formaldehyde for virus inactivation is so low, the concern is only theoretical and does not have relevance for human applications (see Plotkin, Vaccines 2018). This should clearly reported in a Journal devoted to Virologists and immunology experts.

Response 1: The reviewer raised an important fact about the minimal risk pose by low concentration of formaldehyde usage in inactivated vaccine production. Due to this, we decided to remove the text “For example, formaldehyde inactivated RSV and measles vaccines worsened subsequent natural infections [13]. More importantly, formaldehyde is a toxic chemical that may affect the workers during the production of the formaldehyde-inactivated vaccine [14]. Besides that, there is a risk of formaldehyde contamination in the vaccine formulations which raises the concern among the general public regarding the safety of administering multiple doses of formaldehyde-inactivated vaccines in humans [14]” from the manuscript. Please refer to lines 41 to 44 in the revised manuscript.

Point 2: Lines 55 and 57: past tense nont appropriate "discussed" etc.

Response 2: We have amended the text as suggested. “discussed” was changed to “discuss” (line 48 in the revised manuscript) and “suggested” was changed to “suggest” (line 50 in the revised manuscript).

Point 3: Line 71: "are all", remove

Response 3: All was removed from the sentence. Sentence now reads “VP1, VP2 and VP3 are structurally similar to one another, each consisting of eight-stranded β-barrels”. Please refer to lines 64 and 65 in the revised manuscript.

Point 4: Line 295: ref 62 "self-biomineralizing". The term needs clarification. The meaning is NOT obvious, nor well accepted in the literature. Try explaining (if needed).

Response 4: To clarify the term, we have amended the part and now reads “Subsequently, Lyu et. al. (2013) inserted nucleating peptides at the same position to produce a thermostable attenuated EV-A71 vaccine [60]. The insertion of peptides endowed the virus with the capacity to generate an exterior calcium phosphate shell. Intriguingly, the modified surface enabled the virus to exhibit improved thermostability and immunogenicity”. Please refer to lines 281-284 in the revised manuscript.

 Point 5: Line 333: cost, change into costly

Response 5: The text was amended as suggested. The word “cost” was changed to “costly”. Please refer to line 320 in the revised manuscript.

Point 6: Line 346: speices: change into species

Response 6: The word “speices” was changed to “species”. Please refer to line 332 in the revised manuscript.

Point 7: Line 346: inhibition of CPE by 95% - the concept is possibly derived from the ref. Its is however clear that "in vitro" an antibody is either neutralizing (ie not producing CPE) or non-neutralizing (poorly-neutralizing), ie letting the CPE start and proceed to the final death of cells (given enough time and adequate culture conditions, of course!). Thus, inhibiting is not correct. Please modify the sentence. The same applies to lines 466-467. Antibody prevents CPE or protects from infection (thus preventing CPE).

Response 7: The sentences were amended as suggested. The first sentence now reads “Both mAbs were able to protect RD cells from EV-A71 infection” (refer to line 438 and 439). The second sentence now reads “In an in vitro neutralization assay, both mAbs were able to protect RD cells from CPE upon EV-A71 infection” (refer to lines 451 and 452).

Point 8: Line 485: Vero cells

Response 8: Changes was made as suggested. The word “Vero” is now capitalized. Please refer to line 467 in the revised manuscript.

Point 9: Line 571: abbreviations - please put in alphabetic order.

 Response 9: The abbreviations were rearranged in alphabetical order. Please refer to line 553.

Reviewer 2 Report

The review entitled: “Advances in antigenic peptide-based vaccine and 2 neutralizing antibodies against viruses causing Hand, 3 Foot and Mouth Disease” by Anasir and Poh is well structured and well written.

Author Response

Response to Reviewer 2 Comments

 *minor changes were made throughout the text document to correct spellings and improve readability of the manuscript. The changed texts are highlighted in yellow.